# Diffusion Models and Semi-Supervised Learners Benefit Mutually with Few Labels

**Zebin You**[1,2,*] **Yong Zhong**[1,2,*] **Fan Bao**[3]**, Jiacheng Sun**[4]**, Chongxuan Li**[1,2,†] **Jun Zhu**[3]

[1] Gaoling School of Artificial Intelligence, Renmin University of China, Beijing, China
[2] Beijing Key Laboratory of Big Data Management and Analysis Methods, Beijing, China
[3] Dept. of Comp. Sci. & Tech., BNRist Center, THU-Bosch ML Center, Tsinghua University
[4] Huawei Noah's Ark Lab

zebin@ruc.edu.cn; yongzhong@ruc.edu.cn; bf19@mails.tsinghua.edu.cn;
sunjiacheng1@huawei.com; chongxuanli@ruc.edu.cn; dcszj@tsinghua.edu.cn

## Abstract

In an effort to further advance semi-supervised generative and classification tasks, we propose a simple yet effective training strategy called *dual pseudo training* (DPT), built upon strong semi-supervised learners and diffusion models. DPT operates in three stages: training a classifier on partially labeled data to predict pseudo-labels; training a conditional generative model using these pseudo-labels to generate pseudo images; and retraining the classifier with a mix of real and pseudo images. Empirically, DPT consistently achieves SOTA performance of semi-supervised generation and classification across various settings. In particular, with one or two labels per class, DPT achieves a Fréchet Inception Distance (FID) score of 3.08 or 2.52 on ImageNet $256 \times 256$. Besides, DPT outperforms competitive semi-supervised baselines substantially on ImageNet classification tasks, *achieving top-1 accuracies of 59.0 (+2.8), 69.5 (+3.0), and 74.4 (+2.0)* with one, two, or five labels per class, respectively. Notably, our results demonstrate that diffusion can generate realistic images with only a few labels (e.g., $< 0.1\%$) and generative augmentation remains viable for semi-supervised classification. Our code is available at *https://github.com/ML-GSAI/DPT*.

## 1 Introduction

Diffusion probabilistic models [1, 2, 3, 4, 5, 6, 7] have achieved excellent performance in image generation. However, empirical evidence has shown that labeled data is indispensable for training such models [8, 4]. Indeed, lacking labeled data leads to much lower performance of the generative model. For instance, the representative work (i.e., ADM) [4] achieves an FID of 10.94 on fully labeled ImageNet $256 \times 256$, while an FID of 26.21 without labels.

To improve the performance of diffusion models without utilizing labeled data, prior work [8, 9] initially conducts clustering and subsequently trains diffusion models conditioned on the cluster indices. Although these methods can, in some instances, exhibit superior performance over supervised models on low-resolution data, such phenomena have not yet been observed on high-resolution data (e.g., on ImageNet $256 \times 256$, an FID of 5.19, compared to an FID of 3.31 achieved by supervised models, see Appendix C). Besides, cluster indices may not always align with ground truth labels, making it hard to control semantics in samples. Compared to unsupervised methods, semi-supervised generative models [10, 11, 12] often perform much better and provide the same way to control the

---

[*]Equal contribution.
[†]Correspondence to Chongxuan Li.

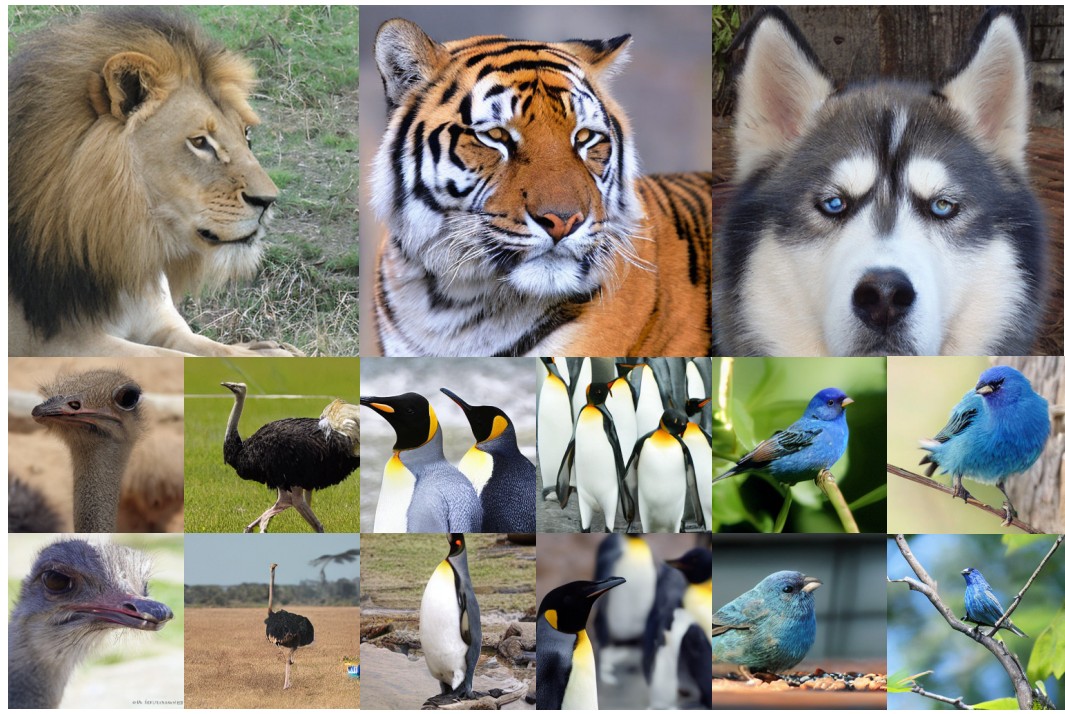

Figure 1: Selected samples from DPT. Top row: $512 \times 512$ samples from DPT trained with **five ($<$ 0.4%)** labels per class. Bottom rows: $256 \times 256$ samples from DPT trained with **one ($<$ 0.1%)** label per class (*Left:* "Ostrich"; *Mid:* "King penguin"; *Right:* "Indigo bunting").

semantics of samples as the supervised ones by using a small number of labels. However, to our knowledge, although it is attractive, little work in the literature has investigated semi-supervised diffusion models. This leads us to a key question: can diffusion models generate high-fidelity images with controllable semantics given only a few (e.g., $< 0.1\%$) labels?

On the other hand, while it is natural to use images sampled from generative models for semi-supervised classification [10, 11], discriminative methods [13, 14, 15] dominant the area recently. In particular, self-supervised based learners [16, 17, 18] have demonstrated state-of-the-art performance on ImageNet. However, generative models have rarely been considered for semi-supervised classification recently. Therefore another key question arises: can generative augmentation be a useful approach for such strong semi-supervised classifiers, with the aid of advanced diffusion models?

To answer the above two key and pressing questions, we propose a simple but effective training strategy called *dual pseudo training* (DPT), built upon strong diffusion models and semi-supervised classifiers. DPT is three-staged (see Fig. 3). First, a classifier is trained on partially labeled data and used to predict pseudo-labels for all data. Second, a conditional generative model is trained on all data with pseudo-labels and used to generate pseudo images given labels. Finally, the classifier is trained on real data augmented by pseudo images with labels. Intuitively, in DPT, the two opposite conditional models (i.e. diffusion model and classifier) provide complementary learning signals to each other and benefit mutually (see a detailed discussion in Appendix E).

We evaluate the effectiveness of DPT through diverse experiments on multi-scale and multi-resolution benchmarks, including CIFAR-10 [19] and ImageNet [20] at resolutions of $128 \times 128$, $256 \times 256$, and $512 \times 512$. Quantitatively, DPT obtains SOTA semi-supervised generation results on two common metrics, including FID [21] and IS [22], in all settings. In particular, in the highly appealing task, i.e. ImageNet $256 \times 256$ generation, DPT with *one* (i.e., $< 0.1\%$) labels per class achieves an FID of 3.08, outperforming strong supervised diffusion models including IDDPM [23], CDM [24], ADM [4] and LDM [25] (see Fig. 2 (a)). It is worth noting that the comparison with previous models here is meant to illustrate that DPT maintains good performance even with minimal labels, rather than directly comparing it to these previous models (direct comparison is unfair as different diffusion models were used). Furthermore, DPT with *two* (i.e., $< 0.2\%$) labels per class is comparable to supervised baseline

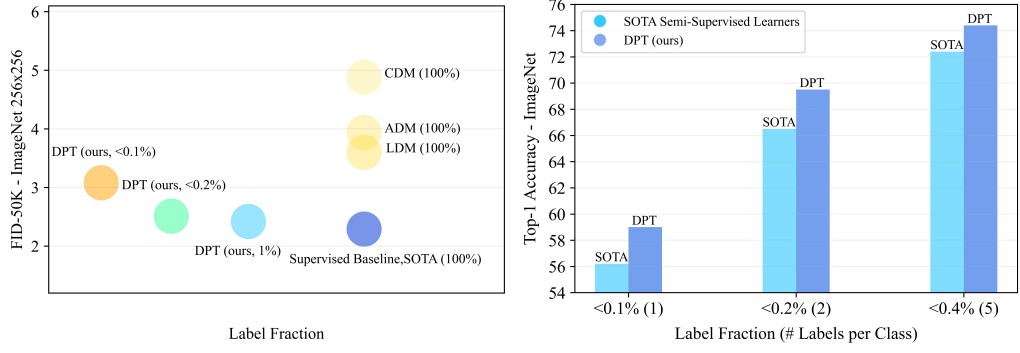

(a) DPT vs. supervised diffusion models.   (b) DPT vs. semi-supervised classifiers.

Figure 2: **Generation and classification results of DPT on ImageNet with few labels.** (a) DPT with $< 0.1\%$ labels outperforms strong supervised diffusion models [4, 24, 25]. (b) DPT substantially improves SOTA semi-supervised learners [17].

U-ViT [5] (FID 2.52 vs. 2.29). Moreover, on ImageNet $128 \times 128$ generation, DPT with *one* (i.e., $< 0.1 \%$) labels per class outperforms SOTA semi-supervised generative models $S^3$GAN [12] with $20\%$ labels (FID 4.59 vs. 7.7). Qualitatively, DPT can generate realistic, diverse, and semantically correct images with very few labels, as shown in Fig 1. We also explore why classifiers can benefit generative models through class-level visualization and analysis in Appendix H.

As for semi-supervised classification, DPT achieves state-of-the-art (SOTA) performance in various settings, including ImageNet with one, two, five labels per class and $1\%$ labels. On the smaller dataset, namely CIFAR-10, DPT with four labels per class achieves the second-best error rate of $4.68_{\pm 0.17}\%$. Besides, on ImageNet classification benchmarks with one, two, five labels per class and $1\%$ labels, DPT outperforms competitive semi-supervised baselines [17, 16], achieving state-of-the-art top-1 accuracy of 59.0 (+2.8), 69.5 (+3.0), 74.4 (+2.0) and 80.2 (+0.8) respectively (see Fig. 2 (b)). Similarly to generation tasks, we also investigate why generative models can benefit classifiers via class-level visualization and analysis in Appendix I.

In summary, our novelty and key contributions are as follows:

- We present Dual Pseudo Training (DPT), a straightforward yet effective strategy designed to advance the frontiers of semi-supervised diffusion models and classifiers.

- We achieve SOTA semi-supervised generation performance on CIFAR-10 and ImageNet datasets across various settings. Moreover, we demonstrate that diffusion models with a few labels (e.g., $< 0.1\%$) can generate realistic, diverse, and semantically accurate images, as depicted in Fig 1.

- We achieve SOTA semi-supervised classification performance on ImageNet datasets across various settings and the second-best results on CIFAR-10. Besides, we demonstrate that aided by diffusion models, generative augmentation remains a viable approach for semi-supervised classification.

- We explore why diffusion models and semi-supervised learners benefit mutually with few labels via class-level visualization and analysis, as showcased in Appendix H and Appendix I.

## 2   Settings and Preliminaries

We present settings and preliminaries on two representative self-supervised based learners for semi-supervised learning [17] [16] in Sec. 2.1 and conditional diffusion probabilistic models [2, 5, 26] in Sec. 2.2, respectively. We consider image generation and classification in semi-supervised learning, where the training set consists of $N$ labeled images $\mathcal{S} = \{(\mathbf{x}_i^l, y_i^l)\}_{i=1}^N$ and $M$ unlabeled images $\mathcal{D} = \{\mathbf{x}_i^u\}_{i=1}^M$. We assume $N \ll M$. For convenience, we denote the set of all real images as $\mathcal{X} = \{\mathbf{x}_i^u\}_{i=1}^M \cup \{\mathbf{x}_i^l\}_{i=1}^N$, and the set of all possible classes as $\mathcal{Y}$.

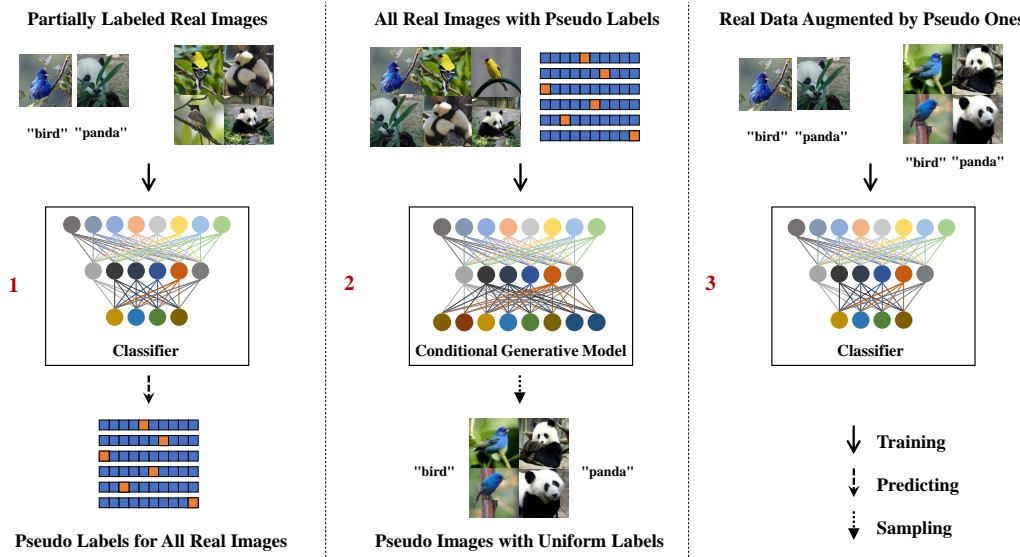

**Figure 3: An overview of DPT**. First, a (semi-supervised) classifier is trained on partially labeled data and used to predict pseudo-labels for all data. Second, a conditional generative model is trained on all data with pseudo-labels and used to generate pseudo images given random labels. Finally, the classifier is trained or fine-tuned on real data augmented by pseudo images with labels.

## 2.1 Semi-Supervised Classifier

**Masked Siamese Networks (MSN)** [17] employ a ViT-based [27] anchor encoder $f_{\boldsymbol{\theta}}(\cdot)$ and a target encoder $f_{\bar{\boldsymbol{\theta}}}(\cdot)$, where $\bar{\boldsymbol{\theta}}$ is the exponential moving average (EMA) [28] of parameters $\boldsymbol{\theta}$. For a real image $\mathbf{x}_i \in \mathcal{X}, 1 \leq i \leq M + N$, MSN obtains $H + 1$ random augmented images, denoted as $\mathbf{x}_{i,h}, 1 \leq h \leq H + 1$. MSN then applies either a random mask or a focal mask to the first $H$ augmented images and obtain $\text{mask}(\mathbf{x}_{i,h}), 1 \leq h \leq H$. MSN optimizes $\boldsymbol{\theta}$ and a learnable matrix of prototypes $\mathbf{q}$ by the following objective function:

$$\frac{1}{H(M+N)} \sum_{i=1}^{M+N} \sum_{h=1}^{H} \text{CE}(\mathbf{p}_{i,h}, \mathbf{p}_{i,H+1}) - \lambda \text{H}(\bar{\mathbf{p}}), \tag{1}$$

where CE and H are cross entropy and entropy respectively, $\mathbf{p}_{i,h} = \text{softmax}((f_{\boldsymbol{\theta}}(\text{mask}(\mathbf{x}_{i,h})) \cdot \mathbf{q}/\tau)$ , $\bar{\mathbf{p}}$ is the mean of $\mathbf{p}_{i,h}$, $\mathbf{p}_{i,H+1} = \text{softmax}(f_{\bar{\boldsymbol{\theta}}}(\mathbf{x}_{i,H+1}) \cdot \mathbf{q}/\tau')$, $\tau, \tau'$ and $\lambda$ are hyper-parameters, and $\cdot$ denotes cosine similarity. MSN is an efficient semi-supervised approach by extracting features for all labeled images in $\mathcal{S}$ and training a linear classifier on top of the features using $L_2$-regularized logistic regression. When a self-supervised pre-trained model is available, MSN demonstrates high efficiency in training a semi-supervised classifier on a single CPU core.

**Semi-ViT** [16] is three-staged. First, it trains a ViT-based encoder $f_{\boldsymbol{\theta}}(\cdot)$ on all images in $\mathcal{X}$ via self-supervised methods such as MAE [29]. Second, $f_{\boldsymbol{\theta}}(\cdot)$ is merely fine-tuned on $\mathcal{S}$ in a supervised manner. Let $\bar{\boldsymbol{\theta}}$ be the EMA of $\boldsymbol{\theta}$, and $\mathbf{x}_i^{u,s}$ and $\mathbf{x}_i^{u,w}$ denote the strong and weak augmented versions of $\mathbf{x}_i^u$ respectively. Finally, Semi-ViT optimizes a weighted sum of two cross-entropy losses:

$$\mathcal{L} = \mathcal{L}_l + \mu \mathcal{L}_u = \frac{1}{N} \sum_{j=1}^{N} \text{CE}(f_{\boldsymbol{\theta}}(\mathbf{x}_j^l), \text{vec}(y_j^l)) +$$

$$\frac{\mu}{M} \sum_{i=1}^{M} \mathbb{I}[f_{\bar{\boldsymbol{\theta}}}(\mathbf{x}_i^{u,w})_{\hat{y}_i} \geq \tau] \text{CE}(f_{\boldsymbol{\theta}}(\mathbf{x}_i^{u,s}), \text{vec}(\hat{y}_i)), \tag{2}$$

where $f_{\bar{\boldsymbol{\theta}}}(\mathbf{x})_y$ is the logit of $f_{\bar{\boldsymbol{\theta}}}(\mathbf{x})$ indexed by $y$, $\hat{y}_i = \arg\max_y f_{\bar{\boldsymbol{\theta}}}(\mathbf{x}_i^{u,w})_y$ is the pseudo-label, $\text{vec}(\cdot)$ returns the one-hot representation, and $\tau$ and $\mu$ are hyper-parameters.

## 2.2 Conditional Diffusion Probabilistic Models

**Denoising Diffusion Probabilistic Model (DDPM)** [2] gradually adds noise $\epsilon \sim \mathcal{N}(\mathbf{0}, \mathbf{I})$ to data $\mathbf{x}_0$ from time $t = 0$ to $t = T$ in the forward process, and progressively removes noise to recover data starting at $\mathbf{x}_T \sim \mathcal{N}(\mathbf{0}, \mathbf{I})$ in the reverse process. It trains a predictor $\epsilon_{\theta}$ to predict the noise $\epsilon$ by the following objective:

$$\mathcal{L} = \mathbb{E}_{t, \mathbf{x}_0, \epsilon}[||\epsilon_{\theta}(\mathbf{x}_t, \mathbf{c}, t) - \epsilon||_2^2], \tag{3}$$

where $\mathbf{c}$ indicates conditions such as classes and texts.

**Classifier-Free Guidance (CFG)** [26] leverages a conditional noise predictor $\epsilon_{\theta}(\mathbf{x}_t, \mathbf{c}, t)$ and an unconditional noise predictor $\epsilon_{\theta}(\mathbf{x}_t, t)$ in inference to improve sample quality and enhance semantics. Formally, CFG iterates the following equation starting at $\mathbf{x}_T$:

$$\mathbf{x}_{t-1} = \frac{1}{\sqrt{\alpha_t}}(\mathbf{x}_t - \frac{\beta_t}{\sqrt{1 - \bar{\alpha}_t}}\tilde{\epsilon}_t) + \sigma_t^2 \mathbf{z}, \tag{4}$$

where $\tilde{\epsilon}_t = (1 + \omega)\epsilon_{\theta}(\mathbf{x}_t, \mathbf{c}, t) - \omega\epsilon_{\theta}(\mathbf{x}_t, t)$, $\omega$ is the guidance strength, $\mathbf{z} \sim \mathcal{N}(\mathbf{0}, \mathbf{I})$, and $\alpha_t$, $\beta_t$, $\bar{\alpha}_t$ and $\sigma_t$ are constants w.r.t. the time $t$.

**U-ViT** [5] is a ViT-based backbone for diffusion probabilistic models, which achieves excellent performance in conditional sampling on large-scale datasets.

## 3 Method

we propose a three-stage strategy called *dual pseudo training (DPT)* to advance semi-supervised generation and classification tasks, illustrated in Fig. 3 and detailed as follows.

### 3.1 First Stage: Train Classifier

DPT trains a semi-supervised classifier on partially labeled data $\mathcal{S} \cup \mathcal{D}$, predicts a pseudo-label $\hat{y}$ of any image $\mathbf{x} \in \mathcal{X}$ by the classifier, and constructs a dataset consisting of all images with pseudo-labels, i.e. $\mathcal{S}_1 = \{(\mathbf{x}, \hat{y}) | \mathbf{x} \in \mathcal{X}\}^3$. Notably, here we treat the classifier as a black box without modifying the training strategy or any hyperparameter. Therefore, any well-trained classifier can be adopted in DPT in a plug-and-play manner. Indeed, we use recent advances in self-supervised based learners for semi-supervised learning, i.e. MSN [17], and Semi-ViT [16]. These two classifiers both provide the generative model with accurate, low-noise labels of high quality.

### 3.2 Second Stage: Classifier Benefits Generative Model

DPT trains a conditional generative model on all real images with pseudo-labels $\mathcal{S}_1$, samples $K$ pseudo images for any class label $y$ after training, and constructs a dataset consisting of pseudo images with uniform labels[4]. We denote the dataset as $\mathcal{S}_2 = \cup_{y \in \mathcal{Y}} \{(\hat{\mathbf{x}}_{i,y}, y)\}_{i=1}^K$, where $\hat{\mathbf{x}}_{i,y}$ is the $i$-th pseudo image for class $y$. Similarly to the classifier, DPT also treats the conditional generative model as a black box. Inspired by the impressive image generation results of diffusion probabilistic models, we take a U-ViT-based [5] denoise diffusion probabilistic model [2] with classifier-free guidance [26] as the conditional generative model. Everything remains the same as the original work (see Sec. 2.2) except that the set of all real images with pseudo-labels $\mathcal{S}_1$ is used for training.

We emphasize that $\mathcal{S}_1$ obtained by the first stage is necessary. In fact, $\mathcal{S}$ is of small size (e.g., one label per class) and not sufficient to train conditional diffusion models. Besides, it is unclear how to leverage unlabeled data to train such models. Built upon efficient and strong semi-supervised approaches [17, 16], $\mathcal{S}_1$ provides useful learning signals (with relatively small noise) to train conditional diffusion models. We present quantitative and qualitative empirical evidence in Fig. 2 (a) and Fig. 1 respectively to affirmatively answer the first key question, namely, diffusion models with a few labels (e.g., $< 0.1\%$) can generate realistic and semantically accurate images.

---

[3] For simplicity, we also use pseudo-labels instead of the ground truth for real labeled data, which are rare and have a small zero-one training loss, making no significant difference.

[4] The prior distribution of $y$ can be estimated on $\mathcal{S}$.

### 3.3 Third Stage: Generative Model Benefits Classifier

**MSN based DPT.** We train the classifier employed in the first stage on real data augmented by $\mathcal{S}_2$ to boost classification performance. For simplicity and efficiency, we freeze the models pre-trained by Eq. (1) in the first stage and replace $\mathcal{S}$ with $\mathcal{S} \cup \mathcal{S}_2$ to train a linear probe in MSN [17]. DPT substantially boosts the classification performance as presented in Fig. 2 (b).

**Semi-ViT based DPT.** We freeze the models, which are pre-trained in a self-supervised manner in the first stage of Semi-ViT, and replace $\mathcal{S}$ with $\mathcal{S} \cup \mathcal{S}_2$ to train a classifier in the third stage of Semi-ViT. We argue that pseudo images can be used in different stages of Semi-ViT and can both boost the classification performance. (see Appendix F.2).

Both consistent improvements provide a positive answer to the second key question, namely, generative augmentation remains a useful approach for semi-supervised classification. Besides, we can leverage the classifier in the third stage to refine the pseudo-labels and train the generative model with one more stage. Although we observe an improvement empirically (see results in Appendix F.3), we focus on the three-stage strategy in the main paper for simplicity and efficiency.

## 4    Related Work

**Semi-Supervised Classification and Generation.** The two tasks are often studied independently. For semi-supervised classification, classical work includes generative approaches based on VAE [10, 30, 31] and GAN [32, 22, 33, 34, 35, 36], and discriminative approaches with confidence regularization [37, 38, 39, 40], consistency regularization [41, 42, 43, 44, 45, 46, 47, 48, 49, 50, 13, 14, 51, 52] and other approaches [53, 54, 55, 56]. Recently, large-scale self-supervised based approaches [18, 28, 57, 17, 16] have made remarkable progress in semi-supervised learning. Besides, semi-supervised conditional image generation is challenging because generative modeling is more complex than prediction. In addition, it is highly nontrivial to design proper regularization when the input label is missing. Existing work is based on VAE [10] or GAN [11, 12], which are limited to low-resolution data (i.e., $\leq 128 \times 128$) and require 10% labels or so to achieve comparable results to supervised baselines.

In comparison, DPT handles both classification and generation tasks in extreme settings with very few labels (e.g., one label per class, $< 0.1\%$ labels). Built upon recent advances in semi-supervised learners and diffusion models, DPT substantially improves the state-of-the-art results in both tasks.

**Pseudo Data and Labels.** We mention additional empirical work on generating pseudo data for supervised learning [58], adversarial robust learning [59, 60], contrastive representation learning [61] and zero-shot learning [62, 63]. Regarding theory, in the context of supervised classification, Zheng et al. [64] have mentioned that when the training dataset size is small, generative data augmentation can improve the learning guarantee at a constant level. This finding can be extended to semi-supervised classification, which is left as future work.

Besides, prior work [65, 66, 67, 8] uses cluster index or instance index as pseudo-labels to improve unsupervised generation results, which are not directly comparable to DPT. With additional few labels, DPT can generate images of much higher quality and directly control the semantics of images with class labels.

**Diffusion Models.** Recently, diffusion probabilistic models [32, 2, 3, 6] achieve remarkable progress in image generation [4, 25, 24, 8, 26, 7], text-to-image generation [68, 69, 70, 25, 71], 3D scene generation [72], image-editing [73, 74, 75], molecular design [76, 77], and semi-supervised medical science [78, 79]. There are learning-free methods [80, 81, 82, 83, 84] and learning-based ones [85, 86] to speed up the sampling process of diffusion models. In particular, we adopt third-order DPM-solver [84], which is a recent learning-free method, for fast sampling. As for the architecture, most diffusion models rely on variants of the U-Net architecture introduced in score-based models [87] while recent work [5] proposes a promising vision transformer for diffusion models, as employed in DPT.

To the best of our knowledge, there has been little research on semi-supervised conditional diffusion models and diffusion-based semi-supervised classification, which are the focus of this paper.

# 5 Experiment

We present the main experimental settings in Sec. 5.1. For more details, please refer to Appendix C. To evaluate the performance of DPT, we compare it with state-of-the-art conditional diffusion models and semi-supervised learners in Sec. 5.2 and Sec. 5.3 respectively. We also visualize and analyze the interaction between the stages to explain the excellent performance of DPT (see Appendix I, H).

## 5.1 Experimental Settings

**Dataset.** We evaluate DPT on the ImageNet [20] dataset, which consists of 1,281,167 training and 50,000 validation images. In the first and third stages, we use the same pre-processing protocol for real images as the baselines [17, 16]. For instance, in MSN, the real data are resized to $256 \times 256$ and then center-cropped to $224 \times 224$. In the second stage, real images are center-cropped to the target resolution following [5]. In the third stage, we consider pseudo images at resolution $256 \times 256$ and center-crop them to $224 \times 224$. For semi-supervised classification, we consider the challenging settings with one, two, five labels per class and 1% labels. The labeled and unlabeled data split is the same as that of corresponding methods [17, 16]. We also evaluate DPT on CIFAR-10 (see detailed experiments in Appendix A).

**Baselines.** For semi-supervised classification, we consider state-of-the-art semi-supervised approaches [17, 16] in the setting of low-shot (e.g., one, two, five labels per class and 1% labels) as baselines. For conditional generation, we consider the state-of-the-art diffusion models with a U-ViT architecture [5] as the baseline.

**Model Architectures and Hyperparameters.** For a fair comparison, we use the exact same architectures and hyperparameters as the baselines [17, 16, 5]. In particular, for MSN based DPT, we use a ViT B/4 (or a ViT L/7) model [17] for classification and a U-ViT-Large (or a U-ViT-Huge) model [5] for conditional generation. As for Semi-ViT based DPT, we use a ViT-Huge model [16] for classification and a U-ViT-Huge model [5] for conditional generation. More details are provided in Appendix C for reference.

**Evaluation metrics.** We use the top-1 accuracy on the validation set to evaluate classification performance. For a comprehensive evaluation of generation performance, we first consider the Fréchet inception distance (FID) [21], sFID [88], Inception Score (IS) [22], precision, and recall [89] on 50K generated samples. We calculate all generation metrics based on the implementation of ADM [4]. We also add the metric $FID_{CLIP}$, which operates similarly to FID but substitutes the Inception-V3 feature spaces with CLIP features, to eliminate confusion that FID can be artificially reduced by aligning the histograms of Top-N classifications without the actual improvement of image quality [90].

**Implementation.** DPT is easy to understand and implement. In particular, it only requires several lines of code based on the implementation of the classifier and conditional diffusion model. We provide the pseudocode of DPT in the style of PyTorch in Appendix B.

**The choice of *K* and *CFG*.** We conduct detailed ablation experiments on the number of augmented pseudo images per class (i.e., $K$) and the classifier-free guidance scale (i.e., $CFG$) in Appendix G and find that the optimal $K$ value is 128 and the optimal $CFG$ values for different ImageNet resolutions are 0.8 for $128 \times 128$, 0.4 for $256 \times 256$, and 0.7 for $512 \times 512$.

**The choice of resolution and number of labels.** We were primarily driven by the task of ImageNet 256×256 generation to systematically compare with a large family of baselines. In this context, we conducted detailed experiments, including settings with one, two, five labels per class, and 1% labels. We find that the performance of DPT with five labels per class is comparable to the supervised baseline, leading us to use this setting as the default in our other tasks such as ImageNet 128×128 and ImageNet 512×512 generation.

## 5.2 Image Generation with Few Labels

We show that diffusion models with a few labels can generate realistic and semantically accurate images. In particular, DPT achieves better results than semi-supervised methods on ImageNet $128 \times 128$ and comparable results to supervised methods on both ImageNet $256 \times 256$ and $512 \times 512$.

Table 1: **Image generation results on ImageNet $128 \times 128$.** $^\dagger$ labels the results taken from the corresponding references and $^\star$ labels baseline achieved by us. We **bold** the best result under the corresponding setting. *With $< 0.1\%$ labels, DPT outperforms strong semi-supervised generative models $S^3GAN$ [12].*

| Method | Model | Label fraction (# labels/class) | FID-50K ↓ | IS ↑ |
|---|---|---|---|---|
| U-ViT-Huge(**supervised baseline**)$^\star$ | Diff. | 100% | 4.53 | 219.8 |
| $S^3$GAN [12]$^\dagger$ | GAN | 5% | 10.4 | 59.6 |
| $S^3$GAN [12]$^\dagger$ | GAN | 10% | 8.0 | 78.7 |
| $S^3$GAN [12]$^\dagger$ | GAN | 20% | 7.7 | 83.1 |
| DPT (**ours**, with U-ViT-Huge and MSN) | Diff. | $< 0.1\%(1)$ | 4.59 | 153.6 |
| DPT (**ours**, with U-ViT-Huge and MSN) | Diff. | $< 0.4\%(5)$ | **4.58** | **210.9** |

Table 2: **Image generation results on ImageNet $256 \times 256$.** $^\dagger$ labels the results taken from the corresponding references and $^\star$ labels baselines achieved by us. DPT and the corresponding baselines employ the same model architectures [5]. *With $< 0.4\%$ labels, DPT outperforms strong conditional generative models with full labels, including CDM [24], ADM [4] and LDM [25].* We **bold** the best result achieved with full labels and underline the best result achieved with few labels. For a fair comparison, we also list the parameters of the diffusion model, including its auxiliary components.

| Method | Model | Label fraction (# labels/class) | FID ↓ | FID$_{CLIP}$ ↓ | sFID ↓ | IS ↑ | Precision ↑ | Recall ↑ | # Params |
|---|---|---|---|---|---|---|---|---|---|
| IC-GAN [67]$^\dagger$ | GAN | 0% | 15.6 | - | 59.0 | - | - | - | - |
| BigGAN-deep [91]$^\dagger$ | GAN | 100% | 6.95 | - | 7.36 | 171.4 | 0.87 | 0.28 | - |
| StyleGAN-XL [92]$^\dagger$ | GAN | 100% | 2.30 | - | **4.02** | 265.12 | 0.78 | 0.53 | - |
| IDDPM [23]$^\dagger$ | Diff. | 100% | 12.26 | - | 5.42 | - | 0.70 | **0.62** | 550M |
| CDM [24]$^\dagger$ | Diff. | 100% | 4.88 | - | - | 158.71 | - | - | - |
| ADM [4]$^\dagger$ | Diff. | 100% | 3.94 | - | 6.14 | 215.84 | 0.83 | 0.53 | 673M |
| LDM-4-G [25]$^\dagger$ | Diff. | 100% | 3.60 | - | - | 247.67 | **0.87** | 0.48 | 455M |
| DiT-XL/2-G [7] $^\dagger$ | Diff. | 100% | **2.27** | - | 4.60 | **278.24** | 0.83 | 0.57 | 675M |
| U-ViT-Large [5]$^\dagger$ | Diff. | 100% | 3.40 | - | 6.63 | 219.94 | 0.83 | 0.52 | 371M |
| *With U-ViT-Large* | | | | | | | | | |
| **Supervised baseline**$^\star$ | Diff. | 100% | 3.31 | 2.39 | 6.68 | 221.61 | 0.83 | 0.53 | 371M |
| **Unsupervised baseline**$^\star$ | Diff. | 0% | 27.99 | 5.40 | 7.03 | 33.86 | 0.60 | 0.62 | 371M |
| DPT (**ours**, with MSN) | Diff. | $< 0.1\%(1)$ | 4.34 | 2.57 | 6.68 | 162.96 | 0.80 | 0.53 | 371M |
| DPT (**ours**, with MSN) | Diff. | $< 0.2\%(2)$ | 3.44 | 2.37 | 6.58 | 199.74 | 0.82 | 0.53 | 371M |
| DPT (**ours**, with MSN) | Diff. | $< 0.4\%(5)$ | 3.37 | 2.35 | 6.71 | 217.53 | 0.83 | 0.52 | 371M |
| DPT (**ours**, with MSN) | Diff. | $1\%(\approx 12)$ | 3.35 | 2.34 | 6.66 | 223.09 | 0.83 | 0.52 | 371M |
| *With U-ViT-Huge* | | | | | | | | | |
| **Supervised baseline**$^\dagger$ | Diff. | 100% | 2.29 | **1.75** | 5.68 | 263.88 | 0.82 | 0.57 | 585M |
| DPT (**ours**, with MSN) | Diff. | $< 0.1\%(1)$ | 3.08 | 1.84 | 5.56 | 201.68 | 0.80 | 0.58 | 585M |
| DPT (**ours**, with MSN) | Diff. | $< 0.2\%(2)$ | 2.52 | 1.81 | 5.49 | 230.34 | 0.81 | 0.57 | 585M |
| DPT (**ours**, with MSN) | Diff. | $< 0.4\%(5)$ | 2.50 | 1.82 | 5.54 | 243.10 | 0.83 | 0.55 | 585M |
| DPT (**ours**, with Semi-ViT) | Diff. | $1\%(\approx 12)$ | 2.42 | 1.77 | 5.48 | 259.93 | 0.82 | 0.56 | 585M |

We evaluate semi-supervised generation performance of DPT on **ImageNet $128 \times 128$**, as shown in Tab. 1. In particular, DPT with only $< 0.1\%$ labels outperforms the SOTA semi-supervised generative model $S^3$GAN [12] with 20% labels (FID 4.59 vs. 7.7), suggesting DPT has superior label efficiency.

In Tab. 2, we compare DPT with state-of-the-art generative models on **ImageNet $256 \times 256$**. We construct highly competitive baselines based on diffusion models with U-ViT-Large [5]. According to Tab. 2, our supervised and unsupervised baselines achieve an FID of 3.31 and 27.99, respectively. Leveraging the pseudo-labels predicted by the strong semi-supervised learner [17], DPT with few labels improves the unconditional baseline significantly and is even comparable to the supervised baseline under all metrics. In particular, *with only two labels* per class, DPT improves the FID of the unsupervised baseline by 24.55 and is comparable to the supervised baseline with a gap of 0.13. Moreover, we also construct more competitive baselines based on U-ViT-Huge to advance DPT. *With one (i.e., $< 0.1\%$) label per class*, our more powerful DPT achieves an FID of 3.08, outperforming strong supervised diffusion models including IDDPM [23], CDM [24], ADM [4] and LDM [25]. Additionally, with 1% labels, DPT achieves an FID of 2.42, comparable to the

Table 3: **Image generation results on ImageNet $512 \times 512$.** [†] labels the results taken from the corresponding references. We **bold** the best result under the corresponding setting.

| Method | Model | Label fraction (# labels/class) | FID-50K ↓ | IS ↑ |
|---|---|---|---|---|
| BigGAN-deep [91][†] | GAN | 100% | 8.43 | 177.90 |
| StyleGAN-XL [92][†] | GAN | 100% | **2.41** | **267.75** |
| ADM [4][†] | Diff. | 100% | 3.85 | 221.72 |
| DiT-XL/2-G [7][†] | Diff. | 100% | 3.04 | 240.82 |
| U-ViT-Huge (**supervised baseline**)[†] | Diff. | 100% | 4.05 | 263.79 |
| DPT (**ours**, with U-ViT-Huge and MSN) | Diff. | $< 0.4\%$(5) | **4.05** | 252.08 |

Table 4: **Top-1 accuracy on the ImageNet validation set with few labels.** [†] labels the results taken from corresponding references, [‡] labels the results taken from Assran et al. [17] and [⋆] labels the baselines reproduced by us. DPT and the corresponding baseline employ exactly the same classifier architectures. *With one, two, five labels per class and 1% labels, DPT improves the state-of-the-art semi-supervised learner [17, 16] consistently and substantially.* We **bold** the best result under the corresponding setting and underline the second-best result.

| Method | Architecture | Top-1 accuracy ↑ given # labels per class (label fraction) | | | |
|---|---|---|---|---|---|
| | | $1(< 0.1\%)$ | $2(< 0.2\%)$ | $5(< 0.5\%)$ | $\approx 12(1\%)$ |
| EMAN [57][†] | ResNet-50 | - | - | - | 63.0 |
| PAWS [51][†] | ResNet-50 | - | - | - | 66.5 |
| BYOL [28][†] | ResNet-200 | - | - | - | 71.2 |
| SimCLRv2 [18][†] | ResNet-152 | - | - | - | 76.6 |
| Semi-ViT [16][†] | ViT-Huge | - | - | - | 80.0 |
| iBOT [93][‡] | ViT-B/16 | $46.1 \pm 0.3$ | $56.2 \pm 0.7$ | $64.7 \pm 0.3$ | - |
| DINO [94][‡] | ViT-B/8 | $45.8 \pm 0.5$ | $55.9 \pm 0.6$ | $64.6 \pm 0.2$ | - |
| MAE [29][‡] | ViT-H/14 | $11.6 \pm 0.4$ | $18.6 \pm 0.2$ | $32.8 \pm 0.2$ | - |
| MSN [17][†] | ViT-B/4 | $54.3 \pm 0.4$ | $64.6 \pm 0.7$ | $72.4 \pm 0.3$ | 75.7 |
| MSN [17][†] | ViT-L/7 | $57.1 \pm 0.6$ | $66.4 \pm 0.6$ | $72.1 \pm 0.2$ | 75.1 |
| MSN (**baseline**)[⋆] | ViT-B/4 | 52.9 | 64.9 | 72.4 | - |
| DPT (**ours**) | ViT-B/4 | 58.6 | **69.5** | **74.4** | - |
| MSN (**baseline**)[⋆] | ViT-L/7 | 56.2 | 66.5 | 72.0 | - |
| DPT (**ours**) | ViT-L/7 | **58.9** | 69.2 | 73.4 | - |
| Semi-ViT (**baseline**)[⋆] | ViT-Huge | - | - | - | 79.4 |
| DPT (**ours**) | ViT-Huge | - | - | - | **80.2** |

state-of-the-art supervised diffusion model [7]. Lastly, DPT with few labels performs comparably to the fully supervised baseline under the $FID_{CLIP}$ metric, which suggests that DPT can generate high-quality samples and does not achieve a lower FID solely due to better Top-N alignment.

We also conduct an experiment on higher resolution (i.e., $512 \times 512$) in Tab. 3, *with five (i.e., $<$ 0.4%) labels*, DPT achieves an FID of 4.05, which is the same as that of the supervised baseline. The above quantitative results demonstrate that DPT can achieve excellent generation performance and label efficiency at diverse resolutions. Qualitatively, as presented in Fig. 1, DPT can generate realistic, diverse, and semantically correct images even with a single label, which agrees with the quantitative results in Tab. 2 and Tab. 3. We provide more samples and failure cases in Appendix F.1 and a detailed class-wise analysis to show how classification helps generation in Appendix H.

Besides, Tab. 5 in Appendix A compares DPT with state-of-the-art generative models on CIFAR-10. DPT achieves competitive performance using only $0.08\%$ labels with EDM [6], which relies on full labels (FID 1.81 vs. 1.79). This result demonstrates the generalizability of DPT on different datasets.

### 5.3 Image Classification with Few Labels

We demonstrate that generative augmentation remains a useful approach for semi-supervised classification aided by diffusion models. In particular, DPT achieves state-of-the-art semi-supervised classification performance on ImageNet datasets across various settings and the second-best results on CIFAR-10.

Tab. 4 compares DPT with state-of-the-art semi-supervised classifiers on the ImageNet validation set with few labels. Specifically, DPT outperforms strong semi-supervised baselines [17, 16] consistently and substantially *with one, two, five labels per class and 1% labels* and achieves state-of-the-art top-1 accuracies of 59.0, 69.5, 74.4 and 80.2, respectively. In particular, with two labels per class, DPT leverages the pseudo images generated by the diffusion model and improves MSN with ViT-B/4 by an accuracy of 4.6%. Besides, we compare the performance of DPT with that of SOTA fully supervised models (as shown in Tab. 12 in Appendix F.2) and find that DPT performs comparably to Inception-v4 [95], using only 1% labels.

Moreover, Tab. 6 in Appendix A compares DPT with state-of-the-art semi-supervised classifiers on CIFAR-10. DPT with four labels per class achieves the second-best error rate of $4.68_{\pm 0.17}\%$.

## 6 Conclusions

This paper presents a simple yet effective training strategy called DPT for conditional image generation and classification in semi-supervised learning. Empirically, we demonstrate that DPT can achieve SOTA semi-supervised generation and classification performance on ImageNet datasets across various settings. DPT probably inspires future work in diffusion models and semi-supervised learning.

**Limitation.** One limitation of DPT is directly using the pseudo images to improve the performance of DPT for its simplicity and effectiveness while we could use pre-trained models like CLIP to filter out noisy image-label pairs that images do not semantically align well with the label. Another limitation pertains to the direct use of pseudo labels. Given our use of classifier-free guidance, we have the flexibility to assign low-confidence pseudo labels to the null token with a high probability, which aids in filtering out noisy pseudo labels.

**Social impact.** We believe that DPT can benefit real-world applications with few labels (e.g., medical analysis). However, the proposed semi-supervised diffusion models may aggravate social issues such as "DeepFakes". The problem can be relieved by automatic detection with machine learning, which is an active research area.

## Acknowledgement

This work was supported by NSF of China (Nos. 62076145); Beijing Outstanding Young Scientist Program (No. BJJWZYJH012019100020098); Major Innovation & Planning Interdisciplinary Platform for the "Double-First Class" Initiative, Renmin University of China; the Fundamental Research Funds for the Central Universities, and the Research Funds of Renmin University of China (No. 22XNKJ13). C. Li was also sponsored by Beijing Nova Program (No. 20220484044).

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
