# OpenReview forum: "Diffusion Models and Semi-Supervised Learners Benefit Mutually with Few Labels"
_NeurIPS.cc/2023/Conference — NeurIPS 2023 spotlight_

### Official Review · Reviewer_JKn2 · 2023-06-29

**Soundness:** 4 excellent
**Presentation:** 3 good
**Contribution:** 4 excellent
**Rating:** 7
**Confidence:** 4

**Summary:**

In this work, the authors propose a novel training strategy, dual pseudo training (DPT), that involves three steps: First, DPT uses state-of-the-art semi-supervised learning methods to predict pseudo-labels on a partially-labeled dataset and diffusion models. Second, DPT trains a conditional generative model with pseudo-labeled data for pseudo-images. Finally, the classifier is re-trained with a dataset which includes both real and pseudo images. DPT surpasses existing strong baseline diffusion models in terms of FID score and semi-supervised learning methods on image classification on CIFAR-10 and ImageNet.

**Strengths:**

* The idea of using Semi-Supervised Learning (SSL) to train diffusion models and then improve SSL is novel. This is analogous to distillation methods that sometimes lead to better student performance than the teacher network.
* The method boosts the performance of both the diffusion model and the discriminative model, becoming the new state-of-the-art method in both fields.
* The writing is clear. I enjoy reading the paper.
* The ablations in the appendix are comprehensive and answered several questions that I have.

**Weaknesses:**

* The baseline diffusion method in this work that leverages full labels is inherently stronger than LDM. Since DPT does not outperform baseline with full supervision, the claim that "with one or two labels per class, DPT ... surpassing strong diffusion models with full labels, such as IDDPM, CDM, ADM, and LDM" is misleading to the readers that DPT, with pseudo-labels, leads to better performance than using full labels.
* Since the authors use a different setting for diffusion method (e.g., with different backbone), the comparison with previous works (e.g., ADM and LDM) is not fair. The authors are encouraged to either use baseline settings from previous methods or migrate their setting to baseline methods to make sure the comparison is fair.
* The discussions of computational efficiency (e.g., in terms of wall clock time) has been omitted. Diffusion models have been known to be slow in the inference process, especially when applied in the pixel space rather than the latent space. Note that this method is still valuable even if it's not particularly compute efficient, as long as the compute required is reasonable, as many semi-supervised learning methods are also less efficient compared to supervised learning.

**Questions:**

Question:
* The method seems to be sensitive to CFG. If I understand it correctly, a CFG of 0 means no guidance. While methods such as stable diffusion typically use a large CFG for an optimal trade-off between diversity and fidelity (7.5 by default, which is equivalent to 6.5 in the notation of this paper), the method uses CFG less than 1. Why is this large discrepancy?

The authors are welcome to address the questions in the weakness section.

**Limitations:**

As mentioned in the weakness section, the discussions of computational efficiency (e.g., in terms of wall clock time) has been omitted. The authors are encouraged to discuss their compute usage and wall clock time as well as the one for their baseline.

---

> ### Author Rebuttal · Authors · 2023-08-09
>
> We thank reviewer JKn2 for the interest and acknowledgement of our contributions and the valuable comments. We respond below to your questions and concerns.
>
> ##  Weakness1: claim may mislead to the readers.
>
> We appreciate your feedback and understand your concern. We will follow your suggestion and try our best to improve the clarity of the writing in the abstract and introduction of the final version to avoid misleading.
>
>
>
> ##  Weakness2: The comparison is not fair.
>
> Thank you for your valuable comment. We emphasize that the most direct comparable supervised baseline of DPT are U-ViT (on ImageNet) and EDM (on CIFAR-10) with full labels. The choice of these baseline for their performances are nearly the state-of-the-art. Besides, for completeness, we also compare DPT with a large family of prior work. Extensive results already demonstrate the effectiveness of DPT. We will try our best to clarify this thing in the abstract and introduction of the final version.
>
>
>
> ##  Weakness3: The discussions of computational efficiency has been omitted.
>
> We appreciate your valuable feedback. We have presented the computational costs for DPT in Tab. 9 for ImageNet $256\times256$ and we discuss the computational efficiency ( which will be added to the Appendix D of our final version) as follows:
>
> **Regarding semi-supervised classification:** The additional time cost over MSN is approximately 201.7%, calculated as $\frac{Generator ~+ ~DPT ~~extra ~~cost}{Classifier}=\frac{5813}{2881}=201.7$%. Although DPT requires nearly twice the training time compared to the MSN baseline, it's still **more time-efficient** than other methods like Triple-GAN [d], which demands at least five times the training time of its classifier. Furthermore, our use of the DPM-Solver, a fast sampler, ensures that the sampling time is minimal, even in pixel space.
>
> **As for semi-supervised generation:** The extra time (the training time for the classifier) over U-ViT(generator baseline) is approximately 50%, calculated as $\frac{Classifier}{Generator}=\frac{2881}{5760}=50$%.
>
> [d] Li C, Xu K, Zhu J, et al. Triple generative adversarial networks[J]. IEEE transactions on pattern analysis and machine intelligence, 2021, 44(12): 9629-9640.
>
>
>
> ##  Q1: Discrepancy in CFG value between stable diffusion and DPT.
>
> Thank you for raising this insightful question. We clarify that the CFG value only depends on the generative component. In our experiments on ImageNet $256\times256$, the supervised baseline U-ViT achieved its best performance with a CFG of 0.4, as measured under the FID metric. We further investigated this by conducting ablation experiments on CFG, and we found that the optimal CFG value for DPT was also 0.4, consistent with U-ViT. Based on the results above, we suggest using the default cfg of 6.5 when intergrating stable diffusion into DPT.

---

> > ### Comment · Reviewer_JKn2 · 2023-08-10
> >
> > The authors have posted an effective rebuttal that resolves my concerns. I still vote for acceptance of this work.

---

> > > ### Author Response · Authors · 2023-08-11
> > >
> > > Thanks again for your valuable comments and acknowledgement of our work. Please let us know if you have any other questions, comments, or concerns. Thank you very much!

---

### Official Review · Reviewer_hJXD · 2023-07-05

**Soundness:** 3 good
**Presentation:** 2 fair
**Contribution:** 3 good
**Rating:** 6
**Confidence:** 4

**Summary:**

To advance semi-supervised generative and classification tasks, this paper proposes a simple yet effective training strategy termed dual pseudo training (DPT). The proposed DPT operates in three stages: training a strong semi-supervised learner to predict pseudo labels, training a strong diffusion model using both real and pseudo labels to generate pseudo images, and retraining the semi-supervised learner with a mix of real and pseudo images. Experimental results show the effectiveness of the proposed DPT on semi-supervised generation and classification across various settings.

**Strengths:**

1. The paper is well-written.
2. The proposed DPT bridging strong semi-supervised learners and diffusion models by pseudo labels and images is technically sound.
3. The idea that diffusion models and semi-supervised learners benefit mutually with few labels is interesting and of practical use.
4. It is impressive that the proposed DPT using 1% labels yields comparable results to the supervised baseline using 100% labels (cf. Table 2).

**Weaknesses:**

1. Does this paper have any technical novelty? It should be highlighted in the abstract and introduction parts.

2. The proposed DPT improves semi-supervised learning models with generative augmentation via diffusion models. Despite substantial performance improvements, the computing overhead and time budget may be big. It would be better to provide a comparative analysis between the proposed method and existing ones.

3. At the second stage of the proposed DPT, pseudo labels may not always align with ground truth labels, making it hard to control semantics in samples. How does the pseudo label accuracy affect the pseudo image fidelity and semantics? Some illustrative examples are preferred.

4. Would the classifier retrained with a mix of real and pseudo images be used to predict pseudo labels as the first stage? The three stages may form a positive cycle to iteratively improve the pseudo label accuracy and pseudo image semantics.

5. Lines 57-58: The exploration on why classifiers can benefit generative models through class-level visualization and analysis is important for the method validation. It would be better to place it in the main text.

6. In Table 1, the semi-supervised generative model S^3GAN was proposed in 2019. Haven't there been any new methods in the last few years?

7. In tables of image generation results, different methods may differ in the model size, leading to unfair comparison. It would be better to show the number of model parameters in these tables.

8. Many recent related works on semi-supervised learning have not been included, discussed, and compared in this paper, e.g., [a-h].

[a] Zhang et al. Flexmatch: Boosting semi-supervised learning with curriculum pseudo labeling. NeurIPS, 2021.

[b] Tang et al. Stochastic Consensus: Enhancing Semi-Supervised Learning with Consistency of Stochastic Classifiers. ECCV, 2022.

[c] Wang et al. Unsupervised Selective Labeling for More Effective Semi-Supervised Learning. ECCV, 2022.

[d] Tang et al. Towards Discovering the Effectiveness of Moderately Confident Samples for Semi-Supervised Learning. CVPR, 2022.

[e] Lim et al. Class-Attentive Diffusion Network for Semi-Supervised Classification. AAAI, 2021.

[f] Gong, S. et al. (2023). Diffusion Model Based Semi-supervised Learning on Brain Hemorrhage Images for Efficient Midline Shift Quantification. In: Frangi, A., de Bruijne, M., Wassermann, D., Navab, N. (eds) Information Processing in Medical Imaging. IPMI 2023. Lecture Notes in Computer Science, vol 13939.

[g] Alshenoudy, A., Sabrowsky-Hirsch, B., Thumfart, S., Giretzlehner, M., Kobler, E. (2023). Semi-supervised Brain Tumor Segmentation Using Diffusion Models. In: Maglogiannis, I., Iliadis, L., MacIntyre, J., Dominguez, M. (eds) Artificial Intelligence Applications and Innovations. AIAI 2023. IFIP Advances in Information and Communication Technology, vol 675.

[h] Chen et al. Debiased Self-Training for Semi-Supervised Learning. NeurIPS, 2022.

**Questions:**

1. Does the "retraining" in the 3-rd stage mean fine-tuning or training from scratch?

2. Why do the strong diffusion models with full labels, such as IDDPM, CDM, ADM, and LDM, perform worse than the proposed DPL with few labels?

How would the proposed DPT perform if full labels are used? It provides the uppper bound.

3. On Line 54, DiT-XL/2-G [7] (FID 2.27) is the SOTA supervised baseline, rather than [5] (FID 3.40). Please carefully check and correct.

5. In Fig. 2 (a), it is no need to indicate the label fraction by the bubble area, as the x-axis has already indicated the label fraction. Different shapes or colors are enough to show the advantage of the proposed DPT.

6. In Table 2, the "bold" and "underline" look a little messy. Some columns of evaluation metrics have no mark while others have many. A better marking scheme should be figured out to make the table easier to understand.

**Limitations:**

Yes.

---

> ### Author Rebuttal · Authors · 2023-08-09
>
> We thank reviewer hJXD for the acknowledgement of our contributions and the valuable comments. We respond below to your questions and concerns.
>
> ##  Weakness1: Regarding technical novelty.
>
> Thank you for your feedback. Our paper's novelty lies in answering the two key questions and in exploring how diffusion models and semi-supervised classifiers can mutually benefit from each other. We will highlight our novelty in the abstract and introduction parts of our final version.
>
> ##  Weakness2: the computing overhead and time budget may be big.
>
> Thank you for highlighting this concern.
>
> **Regarding computing overhead and time budget,** we present the computational costs for DPT in Tab. 9 for ImageNet $256\times256$. The additional time cost over MSN is approximately 201.7%, calculated as $\frac{Generator ~+ ~DPT ~~extra ~~cost}{Classifier}=\frac{5813}{2881}=201.7$%.
>
> **In a comparative analysis between DPT and existing methods**, although DPT requires nearly twice the training time compared to the MSN baseline, it's still **more time-efficient** than other methods like Triple-GAN [d], which demands at least 5 times the training time of its classifier.
>
> **We'll further discuss this in Appendix D of our final version.**
>
> [d] Li C, Xu K, Zhu J, et al. Triple generative adversarial networks[J]. IEEE transactions on pattern analysis and machine intelligence, 2021, 44(12): 9629-9640.
>
> ##  Weakness3: Pseudo label accuracy's effect on image fidelity and semantics
>
> Thank you for your insightful question. We demonstrate illustrative examples in Appendix F.1. If pseudo labels align with ground truth, DPT generates high-quality images (Fig.5 (a)). If not, DPT may generate samples with incorrect semantics (Fig.5 (d), (g)). Nevertheless, as the number of labels increases, the generation performance of DPT improves, yielding more accurate semantics due to more accurate predictions.
>
> ##  Weakness4: Retraining classifier with mixed real and pseudo images for initial pseudo label prediction
>
> We appreciate your insightful question. In Appendix F.3, we explore using a retrained classifier to predict pseudo labels and train the conditional generative model, forming an additional stage. Tab.13 shows that this indeed improve results. However, we focus on the three-stage strategy in this paper for simplicity and efficiency.
>
> ##  Weakness5: Place class-level visualization and analysis in main text.
>
> Thanks for your valuable suggestion. We will follow your suggestion and place these details in the main text of our final version.
>
> ##  Weakness6: Haven't there been any new semi-supervised generative model in the last few years?
>
> Thanks for your valuable comment. In Tab.1, we compare DPT with other semi-supervised methods on ImageNet $\mathbf{128\times128}$. To our knowledge, there is no other literature reporting semi-supervised generation results on this dataset in the last few years.
>
> ##  Weakness7: It would be better to show the number of model parameters in these tables.
>
> We appreciate your valuable suggestion. We will add the comparison of the number of generative model parameters in these tables of our final version.
>
> ##  Weakness8: Lack of inclusion and comparison with some semi-supervised learning works.
>
> We appreciate your pointing out these mentioned works and your valuable suggestion, we will add the discussion of these works in the related work in our final version.
>
> ##  Q1: Does the "retraining" in the 3-rd stage mean fine-tuning or training from scratch?
>
> Thanks for your valuable comment. When we conduct experiments on ImageNet, the classifiers we use are MSN and Semi-ViT. When utilizing MSN, we freeze the pre-trained models and replace $\mathcal{S}$ with $\mathcal{S} \cup \mathcal{S}_2$ to train a classifier from scratch. When utilizing Semi-ViT, we freeze the models, which are pre-trained in a self-supervised manner during the first stage of Semi-ViT, and replace $\mathcal{S}$ with $\mathcal{S} \cup \mathcal{S}_2$ to train a classifier during the third stage of Semi-ViT. We will clarify this in the method section of our final version.
>
> ##  Q2: Why strong diffusion models perform worse than DPT with few labels? How would the proposed DPT perform if full labels are used?
>
> Thank you for your question.
>
> **As for why such diffusion models perform worse than DPT with few labels:** Firstly, the supervised baseline of DPT is U-ViT, which outperforms IDDPM, CDM, ADM, and LDM. Secondly, semi-supervised classifier provides U-ViT with high-quality pseudo labels. Therefore, the performance of DPT is comparable to U-ViT and surpasses IDDPM, CDM, ADM, and LDM.
>
> **Regarding how would DPT perform if full labels are used:** When all labels are available, the second stage of DPT becomes equivalent to training a supervised diffusion model with real labels. This is essentially the same as a supervised conditional baseline. Therefore, based on the discussion above, the upper bound of DPT is the supervised baseline (e.g. 2.29 FID of U-ViT).
>
> We will clarify this in the experiment section of our final version.
>
> ##  Q3: Correct SOTA supervised baseline on Line 54 from [5] to DiT-XL/2-G [7].
>
> We appreciate your feedback. We will correct it in the final version and clarify this in the introduction of our final version to avoid misleading. In our submission, we compared to U-ViT-Huge[5] (FID 2.29),  as shown in the fifth row from the bottom of Tab.2, while U-ViT-Large[5] (FID 3.40), is shown in the ninth row of Tab.2.
>
> ## Q4: no need to indicate the label fraction by the bubble area.
>
> We appreciate your suggestion regarding our graph design. We will follow your suggestion and change our graph in the final version.
>
> ##  Q5: In Table 2, the "bold" and "underline" look a little messy.
>
> Thank you for raising this issue. In the final version of our paper, we will remove the use of “underline” and only introduce “bold” for FID to make the table easier to understand. We appreciate your suggestion.

---

> > ### Comment · Reviewer_hJXD · 2023-08-22
> >
> > Thanks for the effective rebuttal that well addressed my concerns. Hence, I raise my rating from borderline accept to weak accept. I would be glad to see the authors include the posted revisions in the final version.

---

> > > ### Author Response · Authors · 2023-08-22
> > > **Thanks for the update!**
> > >
> > > Dear Reviewer hJXD,
> > >
> > > We are sincerely grateful for your insightful comments and your generous decision to update the rating to 'weak accept'. We highly appreciate it.
> > >
> > > Best regards,
> > >
> > > Authors

---

### Official Review · Reviewer_wQJF · 2023-07-06

**Soundness:** 2 fair
**Presentation:** 3 good
**Contribution:** 2 fair
**Rating:** 6
**Confidence:** 4

**Summary:**

This work presents an interesting interaction between a generative model and a semi-classifier. The pipeline can boost both the generative sample quality and discriminative performance of the semi-classifier in the few label settings. A three-stage pipeline is designed. First, the semi-classifier is trained on a few label data and the semi-label is acquired for all the unlabeled data. Second, the generative model is trained on both label and semi-label data to achieve comparable sample quality as a full-supervised one. At last, the semi-classifier can further improve its performance from the new samples generated by the learned generative model. DPT with two (i.e., < 0.2%) labels per class is comparable to SOTA supervised baseline (FID 2.52 vs. 2.29).

**Strengths:**

- Well written and easy to follow.
- Motivation is clear and the pipeline is simple.
- Good results. DPT with two (i.e., < 0.2%) labels per class is comparable to SOTA supervised baseline (FID 2.52 vs. 2.29).
- Interesting interaction between semi-classifier and diffusion model.

**Weaknesses:**

- 61-63 Specifically, DPT with four labels per class achieves an error rate of 4.53% on 62 CIFAR-10, which even outperforms the performance of SOTA methods on CIFAR-10 63 with 25 labels per class. The Full-flex[1] algorithm has better performance than 4.53% for 4 labels per class on CIFAR-10, which might make this claim incorrect.
- It's interesting to see if the framework is general to other semi-algorithm and diffusion model pairs.
- lack of analysis of the key components of the interaction between the generative model and semi-classifier. For example, what is the key factor to make semi-classifier success for generative model in the design? And What is the key factor to make genertive model success for semi-classifier in the design? This should be discussed in the paper.


[1] Boosting Semi-Supervised Learning by Exploiting All Unlabeled Data

**Questions:**

- Is the algorithm stable for different starting samples in the first stage?
-  what are the key components for the successful interaction between the generative model and semi-classifier?
- How about the PSNR or other commonly used generation evaluation metrics? Since FID and IS are highly imagenet related.
- Can it improve the upper bound of generation quality? If combining full supervised label data with this pipeline, will it surpass the 2.29 FID?
- What about the improvement of unconditional generation?
- The weaknesses mentioned above should be addressed.

**Limitations:**

yes

---

> ### Author Rebuttal · Authors · 2023-08-09
>
> We thank the reviewer wQJF for the valuable and constructive comments. We respond below to your questions and concerns.
>
> ## Weakness1: Claim about DPT with four labels per class may be incorrect.
>
> We appreciate your pointing out the work [c] and your valuable suggestion. DPT with four labels per class achieves an error rate of 4.53% on CIFAR-10, which is comparable to an error rate of 4.44$\pm$0.15 % achieved by Full-flex [c]. We will revise our claim in the introduction and appendix of our final version.
>
> Moreover, Full-flex and our work DPT are orthogonal. As DPT is a flexible framework, integrating Full-flex into DPT could possibly further improve performance.
>
> [c] Boosting Semi-Supervised Learning by Exploiting All Unlabeled Data.
>
> ##  Weakness2: If the framework is general to other semi-algorithm and diffusion model pairs.
>
> Thank you for your valuable question. In principle, DPT can leverage any SSL algorithms or generative models in a plug-and-play manner. In our submission, we have utilized three classical and strong semi-supervised methods including MSN, Semi-ViT, and FreeMatch , as well as two state-of-the-art diffusion models, namely U-ViT (transformer-based) and EDM (U-net-based). Extensive results demonstrate the effectiveness of DPT and suggest that DPT can be generalized to other semi-supervised algorithm and diffusion model pairs.
>
> ##  Weakness3 and Q2: lack of analysis of the key components of the interaction.
>
> We appreciate your feedback. The key component of the interaction between the generative model and semi-classifier is that each can provide high-quality pseudo-data for the other. We explore why classifiers can benefit generative models through class-level visualization and analysis in Appendix H. Besides, we also investigate why generative models can benefit classifiers via class-level visualization and analysis in Appendix I. We will make it clearer in the final version.
>
> ## Q1: Is the algorithm stable for different starting samples in the first stage?
>
> Thank you for your question. We guess that "starting samples" means labeled data in your comment. If we have any misunderstandings, please feel free to post more questions.
>
> The semi-supervised classifiers (i.e. MSN, FreeMatch) we employed in the first stage are stable (for instance, FreeMatch with four labels per class achieves an error rate of $4.90\pm0.04$% on CIFAR-10), thus capable of providing high-quality pseudo-labels to the generative models. In return, the generative models can provide high-quality pseudo-images.
>
> Besides, we're running an experiment by randomly sampling labeled data in stage 1 three times on CIFAR-10, and once the results are available, we will update our response. We will add the discussion and experiments in the final version.
>
> ## Q3: How about the PSNR or other commonly used generation evaluation metrics?
>
> We appreciate your insightful suggestion.
>
> To our best knowledge, calculating PSNR requires comparing two images (e.g. a source image and a super-resolution result) and therefore cannot be adopted to evaluate generative performance directly.
>
> Following your suggestions, we **add a new metric $FID_{CLIP}$[a] to evaluate our method without any model pretrained on ImageNet** in Tab. 1 in our response pdf. $FID_{CLIP}$ is similar to FID but replaces the Inception-V3 feature space with CLIP features. DPT with few labels **performs comparably to the fully supervised baseline under the $FID_{CLIP}$ metric**, which remains to show the effectiveness of DPT. We will include the $FID_{CLIP}$ metric in the final version.
>
> [a] T. Kynkäänniemi, T. Karras, M. Aittala, T. Aila, J. Lehtinen, "The Role of ImageNet Classes in Fréchet Inception Distance", ICLR 2023.
>
> ##  Q4: Can it improve the upper bound of generation quality?
>
> Thanks for your valuable question. When all labels are available, the second stage of DPT becomes equivalent to training a supervised diffusion model with real labels. This is essentially the same as a supervised conditional baseline. Therefore, combining fully supervised labeled data, it will not surpass the baseline (e.g. 2.29 FID of U-ViT).
>
> ## Q5: What about the improvement of unconditional generation?
>
> Thank you for your valuable question. DPT primarily targets class-conditional generation given a few labels, and it approximates the conditional score function rather than the unconditional one. For unconditional generation, you can sample images using labels sampled from the dataset's label distribution, then ignore those labels. This approach maintains image quality and it is equivalent to unconditional generation.
>
> Once again, thank you for your constructive feedback. We'll revise our paper according to your suggestions. We kindly request that you consider raising the score accordingly if we addressed your concerns.

---

> > ### Author Response · Authors · 2023-08-12
> > **Additional experiments on the stability of DPT for different starting samples**
> >
> > Thank you for your question about DPT's stability for different starting samples. We add the experiment by randomly sampling labeled data in stage 1 three times on CIFAR-10 in Rebuttal Table 1.
> >
> > As shown in Rebuttal Table 1, our DPT achieves consistent improvement, which demonstrates that DPT is stable for different starting samples.
> >
> > **Rebuttal Table 1.** Error rates on CIFAR-10 32$\times$32 given 4 labels per class.
> >
> > |           Method           | labeled data 0 | labeled data 1 | labeled data 2 |
> > | :------------------------: | :------------: | :------------: | :------------: |
> > |         FreeMatch          |      4.95      |      5.07      |      4.76      |
> > | DPT with EDM and FreeMatch |      4.53      |      4.91      |      4.60      |

---

> > ### Comment · Reviewer_wQJF · 2023-08-20
> >
> > Dear authors,
> >
> > Thanks for the answer from the rebuttal and the pointer from the appendix, most of my concerns are addressed. I raise my rating to 'weak accept'.
> > Still, the explanation of why the classifier and generator could benefit each other is mostly empirical and results-driven. More detailed analysis or theoretical explanation should make this work more solid and potentially have a greater impact.

---

> > > ### Author Response · Authors · 2023-08-20
> > > **Thanks for the update!**
> > >
> > > Dear Reviewer wQJF,
> > >
> > > We are sincerely grateful for your insightful suggestion and your generous decision to update the rating to 'weak accept'. We will follow your suggestion, and try to add more detailed analysis and theoretical explanation in the future.
> > >
> > > Best regards,
> > >
> > > Authors

---

### Official Review · Reviewer_2c3F · 2023-07-06

**Soundness:** 4 excellent
**Presentation:** 4 excellent
**Contribution:** 3 good
**Rating:** 7
**Confidence:** 3

**Summary:**

This paper proposes to exploit generative diffusion models to benefit semi-supervised learning and vice versa.

Specifically, the proposed training strategy called dual pseudo training (DPT) consists of 3 stages:

(1) Train a semi-supervised learning (SSL) model on labeled and unlabeled data

(2) Train a conditional generative diffusion model using pseudo labels predicted by the SSL model in (1)

(3) Use the generative model in (2) to generate more labeled images to train a classifier

DPT achieves both SOTA performances on image generation and semi-supervised classification tasks on ImageNet and CIFAR-10, demonstrating "Diffusion Models and Semi-Supervised Learners Benefit Mutually with Few Labels".


**Strengths:**

(1) The paper is well-written and easy to follow,

(2) The proposed method is simple but effective. And it can leverage any SSL algorithms or generative models in a plug-and-play manner.

(3) The results are very impressive, both in numbers (SOTA in both classification and generation tasks) and samples (Figure 1).

**Weaknesses:**

(1) Naive combination of SSL and generative models is a straightforward idea; the novelty is limited.



**Questions:**

I do not have questions for this paper.

**Limitations:**

As stated by the author, filtering of pseudo labels in stage 1 and pseudo images in stage 2 should be investigated to reduce noise.

---

> ### Author Rebuttal · Authors · 2023-08-09
>
> We thank reviewer 2c3F for the interest and acknowledgement of our contributions and the valuable comments. We respond below to your questions and concerns.
>
> ## Weakness1: the novelty is limited.
>
> Thank you for your feedback. Our method is simple but effective. The novel contributions of our work lie in answering the two key questions and in exploring how diffusion models and semi-supervised classifiers can mutually benefit from each other. We will elaborate more about our novelty in the abstract and introduction parts of our final version.
>
> ## Limitation1: filtering of pseudo data should be investigated.
>
> We appreciate your insightful suggestion.
>
> **Regarding the filtering of pseudo labels in stage 1**, we will add a discussion about it. For instance, we could set pseudo labels, which have lower confidence, to the unconditional class identifier with a higher probability.
>
> **As for the filtering of pseudo images in stage 2**, we've discussed it in the limitation section of our paper. Employing pre-trained models such as CLIP could help us filter out noisy image-label pairs.
>
> We will elaborate more about the filtering strategy in the appendix section of our paper.

---

### Official Review · Reviewer_kGQr · 2023-07-13

**Soundness:** 3 good
**Presentation:** 3 good
**Contribution:** 2 fair
**Rating:** 7
**Confidence:** 4

**Summary:**

This paper empirically studies the effectiveness of combining diffusion models and semi-supervised classification. First, an image encoder is trained using a self-supervised method, then finetuned with a small fraction of the ground-truth labels. Second, the image classifier assigns pseudo-labels to all unlabeled images in the training dataset which are then used to train a label-conditioned diffusion model. Finally, the diffusion model is used to generate image-label pairs that are used to further train the image classifier from the first step. Multiple experiments on CIFAR10 and ImageNet demonstrate the effectiveness of this approach, both in improving the diffusion model and the image classifier.

**Strengths:**

- The paper demonstrates state-of-the-art results on both semi-supervised image classification and semi-supervised image generation
- The presentation of both the model and the experiments is easy to follow and the text is well written
- Exploring how diffusion models and discriminative models like image classifiers can mutually benefit from one another is interesting and relevant to both academics and practitioners

**Weaknesses:**

1. As shown in [1], the FID can be decreased by aligning the histograms of Top-N classifications without improvements to the image quality. Albeit there exists no perfect method to quantify this effect, it would be insightful to see some investigation into whether the lower FID actually translates to improved image quality or merely better alignment in terms of Top-N classifications.

2. In the introduction, the paper claims that (unsupervised conditional) diffusion models trained with cluster assignments as pseudo-labels still underperform their ground-truth supervised counterparts. As shown in [2] this no longer holds true when the cluster number is set higher than the number of ground-truth classes. It would be interesting to explore how this insight could further benefit the semi-supervised setting, for example by further sub-clustering images with the same pseudo-labels. I consider the paper should be published if they fix their claim about unsupervised diffusion models.


[1] T. Kynkäänniemi, T. Karras, M. Aittala, T. Aila, J. Lehtinen, "The Role of ImageNet Classes in Fréchet Inception Distance", ICLR 2023.
[2] H. Tao, D. W. Zhang, Y. Asano, G. J. Burghouts, C. G. M. Snoek, "Self-Guided Diffusion Models", CVPR 2023.

**Questions:**

The paper reports image generation results on ImageNet for multiple resolutions and differing numbers of ground-truth labels per class. What is the motivation behind the specific choices for the resolution and number of labels combinations? Does the choice of the resolution for the generative model affect the classification results?

**Limitations:**

A potential negative impact of this line of work (semi-supervised generation/classification) is that it could amplify stereotypes where a few examples are taken as representative of the whole class (group). Some discussion on this would benefit the paper.

---

> ### Author Rebuttal · Authors · 2023-08-09
>
> We thank reviewer kGQr for the acknowledgement of our contributions and the valuable comments. We respond below to your questions and concerns.
>
> ## Weakness1: Lower FID translates to image quality improvement or better Top-N alignment?
>
> Thank you for your insightful suggestion and pointing out the work [a]. As shown in [a], the $FID_{CLIP}$ metric can avoid such issues and we **have added the $FID_{CLIP}$ results to Tab.1 in our response pdf** following your suggestion. DPT with few labels performs comparably to the fully supervised baseline under the $FID_{CLIP}$ metric, which suggests that DPT can generate high-quality samples, and does NOT just achieve a lower FID due to better Top-N alignment. Your feedback is highly valuable in helping us provide a more comprehensive evaluation of our model. We will add the $FID_{CLIP}$ metric in Tab. 2 of the final version.
>
> [a] T. Kynkäänniemi, T. Karras, M. Aittala, T. Aila, J. Lehtinen, "The Role of ImageNet Classes in Fréchet Inception Distance", ICLR 2023.
>
> ## Weakness2: The claim about unsupervised diffusion models.
>
> Thank you for your insightful suggestion and pointing out the work [b]. We will fix the claim about unsupervised diffusion models following your suggestion. We discuss the work [b] in details (which will be added to the introduction and related work of our final version) as follows:
>
> (1) As shown in [b], unsupervised diffusion models can outperform supervised baselines when the number of clusters is set higher than the number of ground-truth classes. However, this effect hasn't been confirmed in the context of ImageNet with high resolution (i.e., $\geq 128 \times 128$).
>
> (2) The idea of combining the unsupervised conditional diffusion model approach outlined in [b] with semi-supervised methods is interesting and we believe that this could further improve the performance and it is a promising future work.
>
> [b] H. Tao, D. W. Zhang, Y. Asano, G. J. Burghouts, C. G. M. Snoek, "Self-Guided Diffusion Models", CVPR 2023.
>
> ## Q1: Motivation for choices of resolution and number of labels; resolution for generative model impact on classification
>
> Thanks for the valuable question.
>
> **As for the choice of resolution and number of labels**, we were primarily driven by the task of ImageNet $256 \times 256$ generation to systematically compare with a large family of baselines. In this context, we conducted detailed experiments, including settings with one, two, five labels per class, and 1% labels. We find that the performance of DPT with five labels per class is comparable to the supervised baseline, leading us to use this setting as the default in our other tasks such as ImageNet $128 \times 128$ and ImageNet $512 \times 512$ generation.
>
> **Regarding the impact of resolution choice on the classification results**, we haven't directly investigated this yet. It mainly depends on the quality of the pseudo images sampled from stage 2. Intuitively, a higher resolution may lead to a worse generative performance given a few labels. However, empirically, DPT can generate high quality samples on resolution of 128, 256 and 512.
>
> We will add the above discussion in the experimental section of our final version.
>
> ## Limitations: Potential negative impact
>
> Thanks for your suggestion. We will follow your suggestion and add this discussion to the social impact of our final version.

---

> ### Comment · Reviewer_kGQr · 2023-08-20
>
> Thanks for the rebuttal. It addresses my initial concerns hence I raise my score from Weak Accept to Accept.

---

> > ### Author Response · Authors · 2023-08-20
> > **Thanks for the update!**
> >
> > Dear Reviewer kGQr,
> >
> > Thank you very much for your decision to update the rating to 'accept'. We highly appreciate it.
> >
> > Best regards,
> >
> > Authors

---

### Author Rebuttal · Authors · 2023-08-10

# Summary of the revision

We sincerely thank the reviewers for their valuable comments, which help to further improve the quality of our work. We have addressed the detailed comments, and summarize the main revision which will be updated in our final version as follows:

## New results

- We **add the $FID_{CLIP}$ results in our response PDF (Table 1)**.

## Discussion

- We will add the discussion about the $FID_{CLIP}$ metric.
- We will add the discussion about the choices of resolution and number of labels in Experiment.
- We will add more discussion about the filtering strategy in Appendix.
- We will add the discussion about the upper bound of DPT in Experiment.
- We will add the discussion about computational efficiency and computational overhead in Appendix D.
- We will add the discussion about semi-supervised methods mentioned by Reviewer hJXD.

## Writing

- We will add the $FID_{CLIP}$ metric and the number of generative model parameters in tables about image generation results.
- We will fix the claim about unsupervised diffusion models and the performance of DPT on CIFAR-10.
- We will elaborate more about our novelty in Abstract and Introduction.
- We will place class-level visualization and analysis in the main text.
- We will clarify the comparison with previous works (e.g., ADM and LDM).
- We will remove the use of different bubble areas to indicate the label fraction in Fig.2 (a)
- We will remove the "underline" and only introduce "bold" for FID in Table 2.


We hope you may find the response satisfactory. Please let us know if you have any further feedback.

---

### Decision · Program_Chairs · 2023-09-21

**Decision:**

Accept (spotlight)

**Comment:**

This paper proposes a method to combine the benefits of generative diffusion models and semi-supervised learning, and show this procedure benefits both tasks. The reviewers provided positive views on the paper on several important aspects. The proposed method is a simple but effective approach that improves both diffusion models and classifiers, and achieves state-of-the-art results on semi-supervised image classification and generation. The paper is well-written with comprehensive experiments. There are also some concerns expressed such as that the comparisons with prior work may be unfair due to different model sizes and settings, and the paper omits discussion of computational efficiency. Overall, this is an interesting paper that may spark more discussion in bringing the two domains together.